# The Evolution and Takeoff of the Ecuadorian Economic Groups

Ana Belén Tulcanaza-Prieto [1,*,†] and Manuel Eugenio Morocho-Cayamcela [2,†]

1   Escuela de Negocios, Universidad de Las Américas, UDLA, Vía a Nayón, Quito 170124, Ecuador
2   School of Mathematical and Computational Sciences, Yachay Tech University, Hda. San José s/n y Proyecto Yachay, Urcuquí 100119, Ecuador; mmorocho@yachaytech.edu.ec
*   Correspondence: ana.tulcanaza@udla.edu.ec
†   These authors contributed equally to this work.

**Abstract:** An economic group is a collection of parent and subsidiary corporations that operates as a single economic organism under the same legislature of control. The decisions taken by the economic groups in any country are among the most influential factors that impact its market and the country's economic political scenario. This work studies the impact of the Ecuadorian economic groups from 2015 to 2019, where a historical peak of 300 economic groups was reached. However, the taxes representativeness of the Ecuadorian economic groups remained stable during the same period of analysis. We analyzed the financial and fiscal variables of the Ecuadorian ranking of firms, and detected the following of its economic groups: (i) They are still concentrating wealth despite the implementation of hard government policies to transparent the financial and economic information; (ii) They tend to compete in oligopolistic markets, given that their economic and financial decisions are interconnected with their family firms or consortium groups; (iii) They operate in a behavioral nature that follows a linear association between the total income, total assets, total equity, and total tax collection. We hope this work will serve as a future reference for researchers focused on the economic groups of Ecuador and Latin American countries.

**Keywords:** economic groups; family firms; economic concentration; economic power; Latin America; Ecuador

## 1. Introduction

An economic group is defined as a conglomerate of firms that are grouped together by their financial capital (Navarro 1975). The economic groups of a given country can be articulated in agencies, associations, industry, banks, and commerce, granting them the possibility to be an articulated in blocks as well (Fierro 1991). These groups usually have interest in the majority of economic sectors in a country, creating new fields of investment. Peralta (2015) mentioned that there is a *bourgeois-tripod* in Latin America that integrates the agricultural, industrial, and commercial financial sectors. Marchán (2017) studied the economical behavior of Latin America during the 19th century and introduced the concept of *nation-state* to describe the Republic of Ecuador, where the political strategies have been always focused on the economic elites. The *nation-state* concept coined by the author is justified on the economical and financial integration of Ecuador, mainly based on the idea of an economy open to both exports and local production, enabling the *governability* of the country.

Historically speaking, Acosta (2006) reported that the concentration of economic power in Ecuador began just after the foundation of the Republic in 1830. The author defined the four periods of the economic history of Ecuador as (i) the surplus of the colonial era, (ii) the *primary-exporter* modality period, (iii) the period of industrialization and import substitution, and (iv) the period of modern economy. Most of the Ecuadorian elite groups have been, throughout history, concentrated in Quito, Guayaquil, and Cuenca. The pressure of these economic groups on the government decisions have been visible

across (i) the primary products exportation program (mainly for the coastal region), (ii) the manufacturing project for the north-central Andes region, and (iii) by the law of agricultural and industrial development from 1981. Cueva (1988) explained the economical and political history of Ecuador as the domination of the wealthy groups and the struggle for power between social classes. This domination process granted power to landowners, agro-exporters, and the bourgeoisie, furnishing a permanent economic power to the privileged classes. It is worth noting that the 70% of the Ecuadorian financial institutions that went bankrupt in the financial crisis of 1999 belonged to only 200 persons of the 5 largest economic groups (evincing the power of oligarchies in the country) (El Comercio 2010). Ever since, the inclusion of economic groups in the political and economic decisions of the Ecuadorian government has weakened the state, leading to a successive overthrow of presidents, an unexpected freeze of deposits (*banking holiday*), the devaluation of the local currency (Sucre), and the adoption of the American dollar (USD) as the official currency in 2000.

On the other hand, the economic groups in Latin America have shown small *risk-diversification*, compared to the international standards (Schneider 2013). This investment diversification ranges from the agro-export sector to the private financial sector. Lazzarini et al. (2008) referred that the Latin American economic groups are mostly integrated by families. The author stated that Latin America is living in the capitalism of family ties, which is sustained by the close and intertwined relationships between (i) the economy, (ii) politics, and (iii) the state. This triangular alliance has been managing the national and international market opportunities and local political economies. The study of the economic groups in Latin America intensified in the late 1950s (Garrido and Peres 1996), but its actual importance was boosted in the 1970s due to the protectionism of the economic groups by the industrialization and substitution of imports (Vanoni and Rodríguez 2017). Clear evidence of this phenomena was the several economic concentration studies in Chile (Dahse 1981; Lagos 1960), Colombia (Misas 1975; Silva 1977; Wilches and Rodríguez 2016), Nicaragua (Strachan 1976), and Ecuador (Centro de Estudios y Difusión Social (CEDIS) 1986; Fierro 1991 2019; Minaya 2006; Navarro 1975; Pástor 2015; Solano and Tobar 2017; Tobar and Solano 2017; Tulcanaza 2010; Tulcanaza-Prieto 2018; Tulcanaza-Prieto and Morocho-Cayamcela 2018; Vanoni and Rodríguez 2017). Moreover, the economic groups in Latin America have revealed six standard features: (i) they obey their family's ties, (ii) they are influenced by the inheritance of the colony, (iii) their production is diversified, (iv) they are technology consumers, not producers, (v) they are structured by subsidiaries, and (vi) they are usually intermediaries of multinationals.

In this study, we have empirically examined the financial and fiscal variables of the Ecuadorian economic groups, using the rankings of firms provided by the Servicio de Rentas Internas del Ecuador (SRI) from 2016 to 2020 (data correspond to 2015 to 2019) (Servicio de Rentas Internas del Ecuador 2021). Our results show that the number of Ecuadorian economic groups increased during the period of 2015–2019. However, their contribution on the Ecuadorian fiscal variables remained stable over the period of study. Moreover, we have proven the linear association between (i) the total income, (ii) the total assets, and (iii) the total equity-to-tax collection of the economic groups under study. The database incorporates the most up-to-date data from the Ecuadorian economic groups from the last 5 years. Our results identified that the regulator entities and policymakers are the key actors to establish conditions to avoid the economic concentration in the country. Therefore, their role is to monitor the business mergers or the industry strengthening, using a deeper analysis in the short-, medium-, and long-term to identify the possible collusion and fusion risks of the market.

Our manuscript contributes to the literature since the study period corresponding to 2015–2019 shows that the entry of new economic groups is increasing. Only between 2015 and 2019 were 175 Ecuadorian economic groups incorporated, which indicates that the government regulations have begun to be more transparent in the presentation of economic groups' relationships between the parent company and its subsidiaries, exhibiting that

the business environment is constantly expanding and the main actors in the dynamics of globalization continue to be the large industries and financial groups, which serve as a model for small- and medium-sized enterprises. It should also be noted that the research is timely, given the contribution to the gross domestic product (GDP) of the Ecuadorian economic groups. The trend of analysis of the Ecuadorian economic groups is developed in this article. Basically, with a descriptive approach, we are able to understand how the Ecuadorian economic groups participate in the local environment and contribute by taxes to the national economy.

The remaining of the article is organized as follows: Section 2 presents the literature review regarding the formation of economic groups in Ecuador; Section 3 presents our research methodology; Section 4 reveals the empirical findings and discusses the results; and finally Section 5 highlights the conclusions and offers recommendations and research directions for future researchers.

## 2. Literature Review

### 2.1. The Establishment and Control of the Ecuadorian Economic Concentration

The Ecuadorian market can be understood as the place where the suppliers and demanders of goods and services execute their transactions. This market structure allows a deep analysis of the economic and operational establishment of the industry, stimulating the markets to increase their efficiency. In the Ecuadorian market, the number of suppliers and demanders have determined the degree of concentration of the industrial economy, which is measured by the number of firms and their similarity (Furió and Alonso 2008). Under equal conditions, as the number of firms increases, the market concentration decreases, revealing a negative relationship between both variables. The degree of market concentration is also associated with the business' volume and the number of workers. The business' volume is linked to the market share (i.e., the relationship between the firm's sales or production and the same industry variables), and the number of workers refers to the technical and operational collaborators involved in that business. However, this ratio does not always constitute a good proportion since it also depends on the line of business, the characteristics of the economic activity, and the technological level of goods and services. Table 1 shows the market structure according to the number of suppliers and demanders. The maximum degree of market concentration is the pure monopoly, contrary to the perfect competition, which is an economic structure with several suppliers and demanders.

**Table 1.** Market structure according to the number of suppliers and demanders.

| Number of Demanders | Number of Suppliers | | |
|:---:|:---:|:---:|:---:|
| | **One** | **Few** | **Many** |
| One | Bilateral monopoly | Partial monopsony | Monopsony |
| Few | Partial monopoly | Bilateral oligopoly | Oligopsony |
| Many | Pure monopoly | Oligopoly | Perfect competition |

Source: Own elaboration based on the information obtained from Frank (2005).

Article 334, numeral 1, of the Constitución de la República del Ecuador (2008) states that the government must avoid the concentration or hoarding of the productive resources, reducing the presence of monopolies and oligopolies. Moreover, the *Superintendency of Control of the Power of Market* and the Ley Orgánica de Regulación y Control del Poder de Mercado (2011) determine whether an operation of economic concentration can be created or modified, providing the attribution to deny the concentration transaction or determine its conditions. The economic operators involved in economic concentration operations must inform the superintendency when the volume of total business in Ecuador exceeds the amount established by the board of regulation. The superintendency then determines, using a detailed study, if the concentration is authorized, denied, or subordinated, in order to avoid the overall market affectation.

The analysis of the economic concentration has changed during the last decade. For instance, in the 1970s, the analysis focused on the relationship of power between firms and the establishment of economic groups, whereas in the 2010s, its perspective was focused on the administrative capabilities and the corporate governance of firms and economic groups (Manosalve 2015; Tulcanaza-Prieto et al. 2020). A global economic concentration involves hierarchical capitalism, which studies the relationship between the firm's development, labor market, and capital markets (Schneider 2013). In Latin America, this type of capitalism is predominant, in part due to the fragmentation of labor markets and deficiencies in the educational systems, but also due to the reduced qualification of the workforce, which mitigates the investment in research and development (Tulcanaza-Prieto and Lee 2018).

Conclusively, there are two types of concentration that can be identified, a horizontal and a vertical one. The horizontal concentration is usually called *side* or *wide* concentration, and includes the set of firms that work on the same production stage to scale the operation process, reducing the price of raw materials through a wholesale mechanism (Robinson 1957). On the other hand, firms work in different successive production stages in the vertical concentration, including the value-chain in the transformation of raw material into the final product. However, this integrated process is linked to the establishment and propagation of monopolies, which also generate an upward integration, securing the supply of raw material for entrepreneurs, and a downward integration, providing market stability through production (Tulcanaza-Prieto and Lee 2018).

### 2.2. The Development and Integration of Ecuadorian Economic Groups

The definition of economic groups were developed by the economist and academic Francesco Vito during the Great Depression in 1929 (Vito 1935). This definition was linked with the corporate theory and political economy as an alternative to the classical and neoclassical economic theories (Llosas 2005). In Ecuador, article 5 of the *Reglamento para la Aplicación de la Ley de Régimen Tributario Interno (LORTI)* defined an economic group as the set of individuals and firms, national or foreign, which directly or indirectly own 40% (or more) of the shareholding on other firms (Servicio de Rentas Internas del Ecuador 2015). An economic group can be easily formed when the owner(s) controls several firms, makes financial decisions, and defines the investment policies of the economic surplus (Dahse 1981). Moreover, a group can be structured by a family, friendship, or any other business bond (Leff 1978). It can also be integrated by different companies of diverse economic sectors that only share the administrative and financial control, corporate governance plan, or property management strategies (Anaya 1990). The integration of Ecuadorian economic groups has been impulsed by market failures, such as business' information asymmetry, agency problems, institutional immaturity, or high transaction costs. Therefore, the economic groups act as intermediary institutions to join bidders and demands in the same place, facilitating transactions to organizations and business networks (Chavarín 2011). These intermediary institutions are the response to the economic development strategy driven by the local government (Guriev and Rachinsky 2005; Khanna and Yafeh 2007).

Among the main characteristics of the economic groups, we can distinguish the productive conglomeration, a limited separation of ownership and control between firms, and the transversal integration of the financial sector (Leff 1978). Schneider (2013) discussed that the economic groups influence the institutionality and the political economy of a country, given their technology innovation, skills development, and interaction with the political environment.

#### A Brief History of the Research Efforts on the Ecuadorian Economic Groups

The Ecuadorian economy has been studied since the Republic was established. Several research efforts have found a strong dependence between the Ecuadorian economy and the international market and investments. Regarding the foreign investments, and since family groups have strong links with foreign capital, they have been investing with firms from other countries as well. In addition, the productive branches have been controlled

by few families and several firms, which regulate more than half of the national market (Fierro 1991).

During the 1970s, the Ecuadorian economic concentration was centralized on foreign economic groups, with low national capital and with the oil exportation as its major economic activity. At that time, *Guayaquil* and *Filantrópica* were the two family supergroups, which concentrated the economic and financial decisions of the country and excluded new participation in the business (Navarro 1975). Moreover, the powerful economic classes were the ones that distributed the surplus originated in the centralization and capital concentration (Centro de Estudios y Difusión Social (CEDIS) 1986). During the 1980s, the country experienced a monopolization as a result of the creation of conglomerates, accumulation, vertical and horizontal integrations, and diversification. In the same decade, the Ecuadorian productive capital was grounded in the agricultural sector, specifically in agricultural products and primary products that represented the base of the country's economy. For instance, the *cocoa boom* emerged between the period of 1880 to 1920, while the *banana boom* was exploited during the period of 1948 to 1965, allowing the accumulation of wealth and the appearance of one of the most influential economic groups in Ecuador, *Grupo Noboa*; this was associated with international capitals from transnationals, such as *United Fruit* and *Standard Fruit*. Navarro (1975) is one of the pioneer researchers that studied the behavior of the economic groups in Ecuador. He is recognized for (i) having measured the high levels of economic concentration in Ecuador for the first time, (ii) having recognized the economic groups in Ecuador as family clusters, which control a significant number of firms, (iii) having stated that a small number of families were the main actors in the economic dynamic and economic activities in Ecuador, and (iv) having showed that Ecuadorian firms that appear to be independent behave in the same way as a family economic group when they serve the same shareholder. However, Navarro (1975) dedicated his life to studying the economic concentration based on family groups, but not analyzing their impact on the Ecuadorian economy. Similarly, the *Center of Studies and Social Broadcasting of Ecuador* identified the most important monopolies in the country. They analyzed the levels of concentration and capital centralization of these groups in different branches of the Ecuadorian economy (Centro de Estudios y Difusión Social (CEDIS) 1986). On the other side, Fierro (1991) identified the economic areas where the economic groups are not only consolidated, but also generating oligopolistic or monopolistic opportunities. Fierro (1991) showed that a small number of firms that belonged to a specific economic group concentrated a significant level of the national sales. Other previous studies also confirmed the relationship between the amount of sales of firms and the prevalence of economic concentration (Cañas 2015; EKOS 2012; Pástor 2015; Unda and Bethania 2010).

Centro de Estudios y Difusión Social (CEDIS) (1986) classified the Ecuadorian economic groups as economic elites that control firms and operate in more than one city, mainly located in the provinces of Pichincha, Guayas, and Azuay. It was also proved that when those economic elites are linked to financial institutions, they grant them several advantages, such as the facilitation of obtaining credits with preferential interest rates (Centro de Estudios y Difusión Social (CEDIS) 1986; Fierro 1991). Furthermore, the authors explained that the economic groups in Ecuador have access to the main distribution, transportation, and commercialization chains, evincing a perfect control of the entire production processes inside the country. Solano and Tobar (2017) exposed that the economic groups represented about the 50% of the Ecuadorian GDP during 2015. Moreover, the Ecuadorian economic groups are linked to the management of the national media and banking system; for instance, the largest Ecuadorian bank is part of one of the most important economic groups. Tobar and Solano (2017) showed that the link between the Ecuadorian banking system and the corporate sectors is a key factor in the development of the economic groups and economic concentration. The authors debated that, (i) in the majority of the cases, the economic power groups have obtained financial support from bank allies, leveraging their corporate growth, and (ii) proved the significant positive correlation (at the 1% level) between the credit offered by the financial system, and the

incomes generated by production units in the city of Cuenca. Finally, Fierro (2019) showed that the divorce between financial institutions and investors in the media sector might decrease the conflict of interest between firms.

Conclusively, despite all the legal control and all the technological efforts to stop the formation of economic concentration in Ecuador, the presence of economic groups and important oligopolies is evident in the country.

Therefore, the hypothesis of this study can be summarized as follows.

**Hypothesis 1.** *The number of Ecuadorian economic groups exploited during the period from 2015 to 2019. However, their contribution on the Ecuadorian fiscal variables have remained stable.*

### 3. Data Source and Methodology

The information from the economic groups in Ecuador is available through request to the Ecuadorian tax control entity (Servicio de Rentas Internas del Ecuador 2020). The tax control entity is in charge of compiling and publishing information of this nature, which is considered an effective and transparent tool to trace the trajectory and behavior of the Ecuadorian economic groups.

We analyzed the transcendental financial and fiscal variables from the Ecuadorian economic groups in the following periods:

(a)　2015–2019 (financial and fiscal variables);
(b)　2016–2020 (ranking of the economic groups).

We included the composition and financial behavior of the economic groups, and their contribution to the Ecuadorian macroeconomic variables. The information collected from the register of economic groups, corresponds to the most recent database published by Servicio de Rentas Internas del Ecuador (2021). However, the report from 2017 is not available through the server; thus, the exclusion of this year in our study was beyond our reach. The total sample consists of 1.028 observations of the Ecuadorian economic groups. We also computed the correlation coefficients of the total tax collected by the economic groups, and their association with the total income, total assets, and total equity. We assume a positive Pearson correlation coefficient, given that the independent variables (i.e., total income, total assets, and total equity) have been proved to be linearly positively related with the dependent variable (i.e., tax collected) (Tobar and Solano 2017; Tulcanaza-Prieto 2018).

To prove our hypothesis, the methodology of the study includes (i) descriptive statistics, and (ii) a correlation analysis of the financial and fiscal variables from the Ecuadorian economic groups. All variables employed in this study are described in Table 2.

**Table 2.** Financial and fiscal variables of the study.

| Variables | Formula / Description |
| --- | --- |
| Total Assets | Liabilities + Total Equity |
| Total Equity | Total Assets − Liabilities |
| Total Income | Revenue − Cost of goods sold |
| Income Tax | (Taxable base [a] − returns − discounts − costs − all deductions) * 25% |
| Tax Burden | (Income Tax/Total Income) * 100 |
| Total National Net Tax Collection | Total Tax Collection - credit notes − compensations − returns |
| Total Tax Collection | Sum of all Ecuadorian taxes [b] |

Note: [a] composed by ordinary and extraordinary taxed income, [b] includes: (1) income tax collected, (2) value added tax, (3) tax on special consumption, (4) environmental promotion tax, (5) motor vehicle tax, (6) currency outflow tax, (7) abroad assets tax, (8) RISE, (9) royalties, patents, and mining conservation profits, (10) contribution for comprehensive cancer care, (11) one-time and temporary contribution, (12) interest for tax delay, (13) tax fines, and (14) other income. Source: Own elaboration based on Servicio de Rentas Internas del Ecuador (2020).

We have included the following analysis to verify our hypothesis:

(a)　A graphic analysis of the evolution of the Ecuadorian economic groups, and the evolution of the most representative macroeconomic and fiscal variables of Ecuador.

This analysis establishes the representativeness of the Ecuadorian economic groups in the local economy.

(b)  A comparative analysis of the composition of the Ecuadorian economic groups, and the evolution of their financial and fiscal variables. This analysis compares figures from 2015 to 2019 to determine if the Ecuadorian economic groups have increased over time.

(c)  A descriptive analysis of the tax burden of the Ecuadorian economic groups to showcase the amount of paid taxes (considered as a proportion of the total income) in a specified period.

(d)  An analysis of the evolution of the top-10 Ecuadorian economic groups (according to their size and tax collection). This analysis explains the tax representativeness of the top 10 Ecuadorian economic groups vs. all the remaining economic groups and compared the outcome with the total national net tax collection. Moreover, the analysis of the Ecuadorian economic groups includes the contribution on the national net tax collection to verify if their representativeness have remained stable over time.

(e)  A correlation analysis between the financial variables of the Ecuadorian economic groups, and the total tax collection to prove the linear association between variables.

## 4. Results and Discussion

Figure 1 illustrates the evolution of the Ecuadorian economic groups from 2007 to 2020. Specifically, in the period of our analysis, we found an increase of 140.0% of the number of economic groups in Ecuador, raising from 125 (2015) to 300 (2020) economic groups. The provinces of Pichincha and Guayas concentrated an average of 79.3% of economic groups from 2015 to 2020, which denotes the poles of concentration in the provinces with the highest economic growth in Ecuador. The Andes region (provinces of Azuay, Imbabura, Loja, Pichincha, and Tungurahua) showed the highest agglutination of economic groups (on average 62.7%), while the economic groups located in the coastal provinces (Guayas and Manabí) represented an average of 37.2%. The province of Orellana in the Amazon region showed the presence of one economic group during the period of 2019–2020. Moreover, from 2015 to 2016, we found the most significant increase in the number of economic groups, growing from 125 to 200. Vanoni and Rodríguez (2017) explained the raise of 60.0% in the number of economic groups by the implementation of most efficient business strategies.

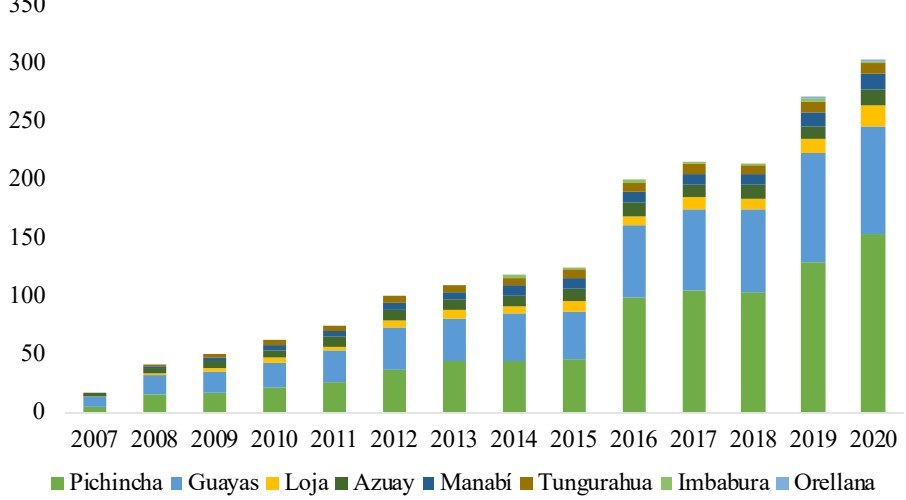

**Figure 1.** Evolution of the Ecuadorian Economic Groups. Source: Own elaboration based on the data collected from Servicio de Rentas Internas del Ecuador (2020).

Figure 2 shows us the most important macroeconomic and fiscal variables from 2015 to 2019 (excluding 2017 given the absence of information on the data source). The total income of economic groups represented on average 63.0% of the gross domestic product

(GDP) from 2015 to 2019. Likewise, the total national net tax collection, and the total tax collection of economic groups, implied on average 12.1%, and 5.9% of the Ecuadorian GDP, respectively. The total tax collection of economic groups represented on average the 48.5% of the total national net tax collection.

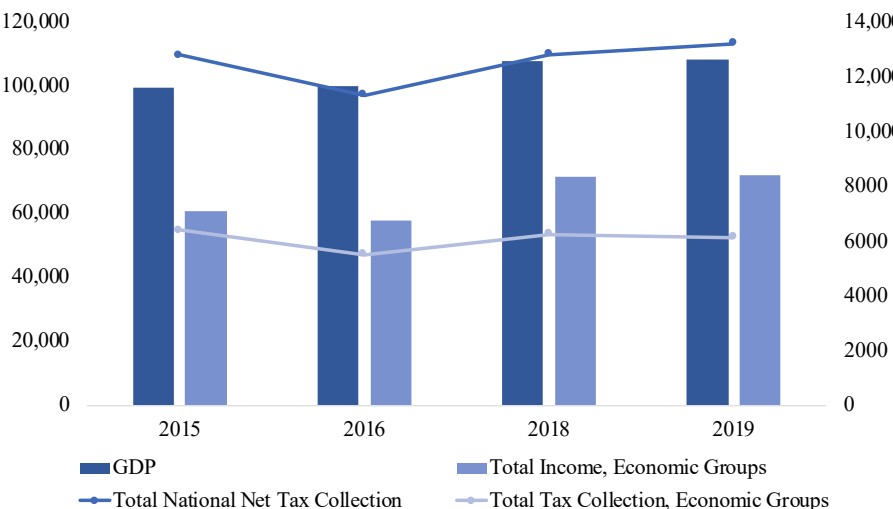

**Figure 2.** Evolution of macroeconomic and fiscal variables (measured in USD millions). Note: Information from 2017 was not available on the data source. Source: Own elaboration based on the data obtained from Servicio de Rentas Internas del Ecuador (2020) and Banco Central del Ecuador (2021).

Table 3 displays the composition of the Ecuadorian economic groups using rankings from 2017 to 2020, as provided by the Servicio de Rentas Internas del Ecuador (2021). Note that in 2017, there were 215 Ecuadorian economic groups with 7126 members; which were integrated by 84.4% of national and foreign firms, and 15.6% from natural and foreign individuals. The economic groups increased by 39.5% during 2020, which means there were 300 economic groups, which were integrated by 9.121 members, represented by 7712 national and foreign firms (84.6% of members of economic groups) and 1409 natural and foreign individuals (15.4% of total members). Moreover, 433 and 453 members were located in *tax havens* during 2017 and 2020, respectively, while 307 and 393 members of economic groups were related to off-shore firms (i.e., *Panama Papers* records) in the same period. Tax havens are linked to the international financial system and globalization (Iturralde 2017). Tax havens are also associated with the expansion and concentration of economic groups, which increase their economic and political power by the evasion or elusion of taxes, moving their capitals to countries or cities with tax havens. The economic group Juan Eljuri is the most visible group associated with tax havens and offshore firms, which represented 9.9% and 10.7% of the total economic groups domiciled in tax havens and offshore firms, respectively (Servicio de Rentas Internas del Ecuador 2020). Furthermore, the number of members of economic groups related to financial institutions increased from 15 to 46 from 2017 to 2020, while there was an increase of 17.0% in the number of members associated with media entities. Navarro (2006) advocated that the media is a *factual power*, given that it provides stability to the government. This factual power has influence in economic power, political permanency, and social response. Therefore, he concluded that the media and financial institutions might be considered a new economic group.

**Table 3.** Composition of Ecuadorian economic groups.

| Detail | Ranking 2017 | Ranking 2020 |
|---|---|---|
| Number of economic groups | 215 | 300 |
| Number of members of the economic group | 7126 | 9121 |
| Number of members domiciled in tax havens | 433 | 453 |
| Number of members as offshore firms (Panama Papers) [a] | 307 | 393 |
| Number of members related with financial institutions | 15 | 46 |
| Number of members related with media entities | 47 | 55 |

Note: [a] Panama Papers are related to members of economic groups which have been identified in the records of Panama Papers, website: https://panamapapers.icij.org Source: Own elaboration based on the information provided by the Servicio de Rentas Internas del Ecuador (2020) .

Table 4 exhibits the evolution of the financial and fiscal variables of the Ecuadorian economic groups from 2015 to 2019. The income tax grew 27.5%; however, the total tax collection decreased in 4.3%, varying from USD 6.394 million (2015) to USD 6.121 million (2019). The total income, total assets, and total equity increased by 17.8%, 36.8%, and 37.0%, respectively. The effective tax rate, computed by the ratio between income tax and total income, was 2.3% in 2015 and 2.5% in 2019, meaning that for every USD 100 that the Ecuadorian economic groups earned, they paid around USD 2.3 and USD 2.5 in 2015 and 2019, respectively.

**Table 4.** Financial and fiscal variables of Ecuadorian economic groups (in USD millions).

| Detail | 2015 | 2019 | Variation | |
|---|---|---|---|---|
| Income Tax | 1389 | 1772 | 382 | 27.5% |
| Total Tax Collection | 6394 | 6121 | −273 | −4.3% |
| Total Income | 60,903 | 71,744 | 10,841 | 17.8% |
| Total Assets | 95,214 | 130,262 | 35,048 | 36.8% |
| Total Debt | N.A. | 1234 | 1234 | |
| Total Equity | 35,206 | 48,216 | 13,009 | 37.0% |

Note: Total debt is not reported for 2015 (N.A.). Source: Own elaboration based on Servicio de Rentas Internas del Ecuador (2020).

Table 5 and Figure 3 portray the tax burden of the Ecuadorian economic groups. The tax burden is calculated by the ratio between the income tax and the total income of each economic group. The tax burden showed a decrease of 5.1%, changing from 2.42 (2015) to 2.29 (2019). Its median value decreased by 10.5%, while its standard deviation value increased by 0.34 units, showing more dispersion and risk in the distribution of the tax burden during 2019 compared to 2015. The tax burden heavily depends on the productive activity, profit (depending on the economic sector), and the taxes according to the firm size and industry. Therefore, the tax burden does not necessarily show an increase in the profit of firms, but rather, it represents the productive stage of economic groups (Revista Líderes 2017). The highest tax burden in 2015 is 9.56 units represented by Grupo Degfer, while Nuevo Rancho Nuransa showed the highest tax burden in 2019 with 16.17 units.

**Table 5.** Descriptive statistics of tax burden of Ecuadorian economic groups.

| Year | Average Tax Burden | Median | Standard Deviation | Maximum |
|---|---|---|---|---|
| 2015 | 2.42 | 1.83 | 1.78 | 9.56 |
| 2019 | 2.29 | 1.64 | 2.11 | 16.17 |

Source: Own elaboration based on the data from Servicio de Rentas Internas del Ecuador (2020).

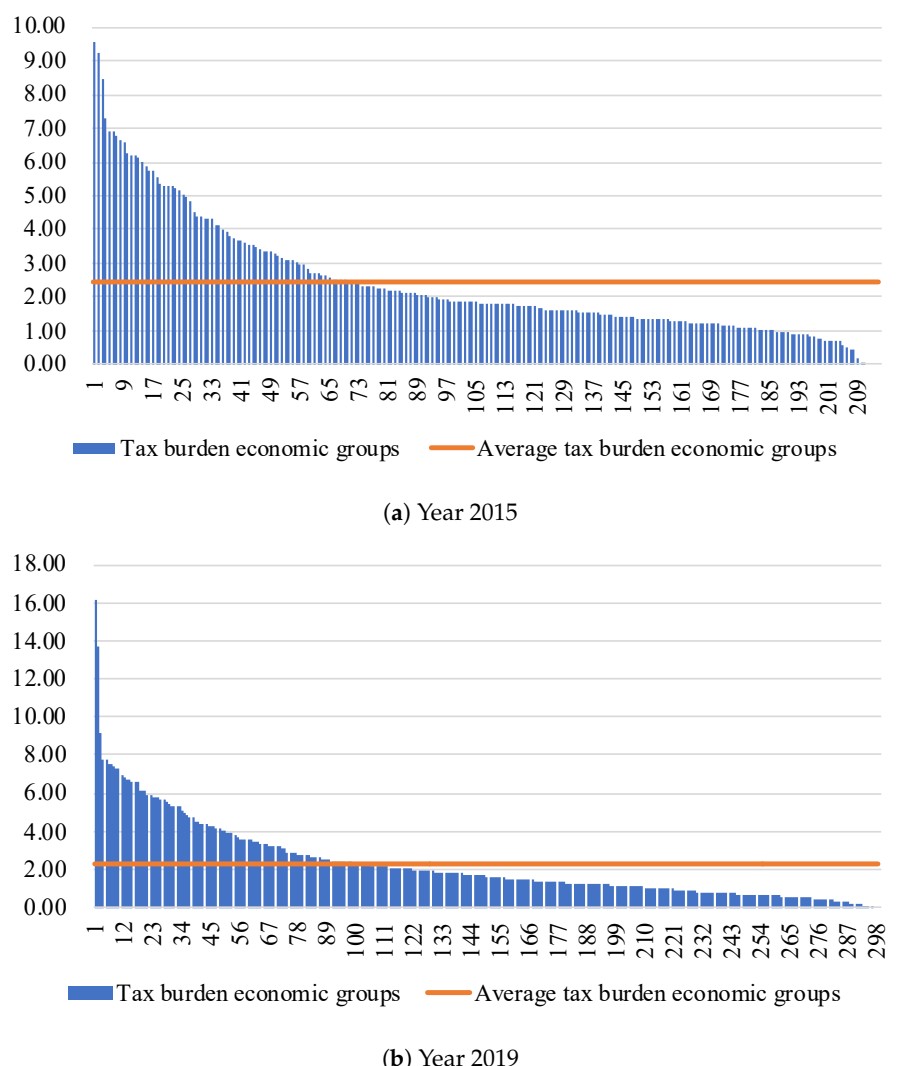

**Figure 3.** Tax burden of Ecuadorian economic groups related to the income tax from the years (**a**) 2015, and (**b**) 2019. Source: Own elaboration based on Servicio de Rentas Internas del Ecuador (2020).

Table 6 shows the top 10 Ecuadorian economic groups classified by their size. Six out of the ten largest economic groups have maintained their position in the top 10 when comparing 2016 to 2020. The total asset of the top ten economic groups in 2020 were USD 53.145 million, with a few financial institutions (Banco Pichincha, Banco de Guayaquil, Produbanco, and Banco Bolivariano CA) concentrating the 60.9% of the total assets of the top 10 ranking.

Table 7 presents the top 10 Ecuadorian economic groups classified by their tax collection using the ranking 2016–2020. Eight out of the ten economic groups have maintained their position in the top ten ranking, comparing 2016 to 2020. If we compare Tables 6 and 7, we can notice that five economic groups were present in both rankings (classified by size and tax collection) during 2020. The total tax collection of the top 10 economic groups represented 43.8% and 44.7% of the total tax collection of the Ecuadorian economic groups for 2015 and 2019, respectively. What is more, the *representativeness* of the total tax collection of the top 10 economic groups over the total national net tax collection varied from 21.9% to 20.8% from 2015 to 2019. Ultimately, the total tax collection of economic groups represented at least half of the total national net tax collection.

**Table 6.** Top 10 Ecuadorian economic groups per year, classification according to their size.

| Rk. 2020 | Rk. 2019 | Rk. 2017 | Rk. 2016 | Variation 2016–2020 | Economic Group |
|---|---|---|---|---|---|
| 1 | 1 | 1 | 1 | 0 | Banco Pichincha |
| 2 | 2 | 3 | 2 | 0 | Almacenes Juan Eljuri |
| 3 | 3 | 4 | 4 | 1 | Corporación Favorita |
| 4 | 4 | 2 | 5 | 1 | Schlumberger del Ecuador |
| 5 | 7 | 7 | 8 | 3 | Banco de Guayaquil |
| 6 | 6 | 6 | 7 | 1 | Produbanco |
| 7 | 5 | 5 | 3 | −4 | OCP Ecuador |
| 8 | 8 | 10 | 11 | 3 | Holdingdine Corporación Industrial y Comercial |
| 9 | 9 | 11 | 10 | 1 | Corporación El Rosado |
| 10 | 10 | 12 | 12 | 2 | Banco Bolivariano C.A. |
| 11 | 12 | 8 | 6 | −5 | Claro |
| 17 | 17 | 9 | 9 | −8 | Industria Pronaca |

Note: Ranking of 2018, which contains information of 2017, was not available on the SRI webpage. Source: Own elaboration based on Servicio de Rentas Internas del Ecuador (2020).

Table 8 shows the principal financial variables of the ranking between 2016 and 2020 of the top 10 Ecuadorian economic groups classified by their size. Among our findings, it is worth mentioning that (i) the income contribution of the ten largest economic groups to the total income of all Ecuadorian groups was 25.5% and 22.4% in 2015 and 2019, respectively; (ii) the total assets grew by 35.5% and 36.8% for the top ten economic groups and total economic groups from 2015 to 2019, respectively; (iii) the total assets of the top ten economic groups represented 41.3% and 40.8% of the total assets of total economic groups during 2015 and 2019, respectively; and the total equity of the ten largest economic groups varied in 38.0%, increasing from USD 11.701 million in 2015 to USD 16.145 million in 2019. The increase represents 33.2% (2015) and 33.5% (2019) when compared to the total equity of the economic groups.

Lastly, Table 9 offers the linear correlation coefficients of the most important financial variables and the total tax collection. Total income, total assets, and total equity revealed a significant positive association at the 1% level with the total tax collection. The highest correlation coefficient is obtained between the total income and the total tax collection, showing coefficient values of 0.808, 0.784, 0.837, and 0.793 for 2015, 2016, 2018, and 2019, respectively. Most of the correlation values themselves were higher than 0.700, which means that the *multicollinearity* problem might arise in the regression analysis, given that the dependent variable is strongly associated with all independent variables, verifying one more time that the total tax collection directly depends on the total income, total assets, and total equity.

**Table 7.** Top 10 Ecuadorian economic groups per year, classification according to tax collection.

| Ranking 2020 | Ranking 2019 | Ranking 2017 | Ranking 2016 | Variation 2016–2020 | Economic Group | 2015 | 2016 | 2018 | 2019 |
|---|---|---|---|---|---|---|---|---|---|
| 1 | 1 | 1 | 1 | 0 | Banco Pichincha | 664,261,405 | 498,739,821 | 607,395,054 | 667,872,343 |
| 2 | 2 | 2 | 2 | 0 | Dinadec | 350,974,788 | 352,100,364 | 390,791,027 | 427,107,715 |
| 3 | 3 | 3 | 4 | 1 | Produbanco | 264,294,993 | 222,306,816 | 274,261,174 | 299,941,296 |
| 4 | 5 | 7 | 8 | 4 | Banco de Guayaquil | 191,614,215 | 175,517,131 | 222,582,890 | 234,359,513 |
| 5 | 6 | 6 | 7 | 2 | Banco Bolivariano C.A. | 197,354,449 | 180,440,219 | 206,430,145 | 209,149,340 |
| 6 | 7 | 5 | 3 | −3 | Claro | 315,489,121 | 185,073,552 | 186,042,898 | 190,495,917 |
| 7 | 4 | 9 | 9 | 2 | Almacenes Juan Eljuri | 183,909,249 | 157,333,985 | 229,738,403 | 190,328,750 |
| 8 | 8 | | | −8 | Banco Internacional | 185,639,529 | 182,165,876 | | |
| 9 | 9 | | | −9 | Arca Ecuador | 183,298,823 | 168,714,088 | | |
| 10 | 10 | 10 | 10 | 0 | Citibank N. A., Ecuador | 179,366,431 | 144,775,146 | 179,592,586 | 166,635,702 |
| | | 4 | 5 | 5 | Itabsa | 239,723,714 | 205,429,092 | | |
| | | 8 | 6 | 6 | Schlumberger dl Ecuador | 210,851,012 | 157,866,766 | | |
| (a) Total Tax Collection, Top 10 Economic Groups | | | | | | 2,797,839,378 | 2,279,582,892 | 2,665,772,529 | 2,736,770,540 |
| (b) Total Tax Collection, Economic Groups | | | | | | 6,393,835,744 | 5,499,929,764 | 6,256,788,523 | 6,120,831,840 |
| (c) Total National Net Tax Collection | | | | | | 12,755,076,181 | 11,309,307,282 | 12,809,502,107 | 13,180,846,182 |
| (a)/(b) | | | | | | 43.8% | 41.4% | 42.6% | 44.7% |
| (a)/(c) | | | | | | 21.9% | 20.2% | 20.8% | 20.8% |
| (b)/(c) | | | | | | 50.1% | 48.6% | 48.8% | 46.4% |

Note: Ranking of 2018, which contains information of 2017, was not available on the SRI webpage. Source: Own elaboration based on Servicio de Rentas Internas del Ecuador (2020).

**Table 8.** Financial variables, top 10 Ecuadorian economic groups per year, classification according to their size (in USD millions).

| Economic Group | Total Income | | | | Total Assets | | | | Total Equity | | | |
|---|---|---|---|---|---|---|---|---|---|---|---|---|
| | 2015 | 2016 | 2018 | 2019 | 2015 | 2016 | 2018 | 2019 | 2015 | 2016 | 2018 | 2019 |
| Banco Pichincha | 2055 | 2102 | 2387 | 2727 | 13,203 | 14,484 | 16,580 | 17,892 | 2658 | 2601 | 3477 | 3681 |
| Almacenes Juan Eljuri | 1894 | 1813 | 3106 | 2740 | 4294 | 4575 | 6506 | 6418 | 1244 | 1287 | 2035 | 1947 |
| Corporación Favorita | 2697 | 2508 | 2880 | 2919 | 2132 | 2245 | 2867 | 3344 | 1567 | 1690 | 2154 | 2315 |
| Schlumberger del Ecuador | 1297 | 2027 | 1724 | 1834 | 3137 | 4537 | 3968 | 3617 | 1546 | 1966 | 2397 | 2306 |
| Banco de Guayaquil | 484 | 467 | 525 | 602 | 3840 | 4190 | 4570 | 5334 | 644 | 670 | 727 | 792 |
| Produbanco | 342 | 357 | 533 | 609 | 3905 | 4324 | 4935 | 5337 | 617 | 623 | 539 | 571 |
| OCP Ecuador | 1877 | 1661 | 2013 | 1579 | 3720 | 3749 | 3771 | 3212 | 1593 | 1455 | 2004 | 1455 |
| Holdingdine | | 1019 | 965 | 1030 | | 2125 | 2185 | 2325 | | 1693 | 1889 | 2031 |
| Corporación El Rosado | 1591 | | 1605 | 1657 | 1635 | | 1655 | 1838 | 520 | | 454 | 477 |
| Banco Bolivariano C.A. | | 289 | 342 | | | 3565 | 3828 | | | 523 | 571 | |
| Claro | 1542 | 1447 | | | 1969 | 1968 | | | 521 | 555 | | |
| Industria Pronaca | 1754 | 1655 | | | 1469 | 1523 | | | 791 | 845 | | |
| (a) Total Top 10 | 15,533 | 15,057 | 16,027 | 16,039 | 39,304 | 43,720 | 50,603 | 53,145 | 11,701 | 13,386 | 16,199 | 16,145 |
| (b) Total Economic Groups | 60,903 | 57,994 | 71,455 | 71,744 | 95,214 | 102,044 | 122,032 | 130,262 | 35,206 | 36,479 | 46,525 | 48,216 |
| (a)/(b) | 25.5% | 26.0% | 22.4% | 22.4% | 41.3% | 42.8% | 41.5% | 40.8% | 33.2% | 36.7% | 34.8% | 33.5% |

Note: Ranking of 2018, which contains information of 2017, was not available on the SRI webpage. Source: Own elaboration based on Servicio de Rentas Internas del Ecuador (2020).

**Table 9.** Pearson correlation coefficients, Ecuadorian economic groups.

| Financial Variables vs. Total Tax Collection | 2015 | 2016 | 2018 | 2019 |
|---|---|---|---|---|
| Total Income | 0.808 *** | 0.784 *** | 0.837 *** | 0.793 *** |
| Total Assets | 0.537 *** | 0.544 *** | 0.736 *** | 0.698 *** |
| Total Equity | 0.677 *** | 0.716 *** | 0.802 *** | 0.772 *** |

Note: Information of 2017 was not available in the SRI webpage, *** indicates statistical significance at the 1% level. Source: Own elaboration based on Servicio de Rentas Internas del Ecuador (2020).

## 5. Conclusions

We have seen how a small cluster of family groups controls the economic power in Ecuador. Those clusters, also called economic groups, have influenced the Ecuadorian market and politics, in part due to their economic power, but also due to their financial, communicational, and political concentration. We have verified that the economic groups in Ecuador manage higher concentrations of wealth despite the implementation of government policies for transparency of the financial and economic information of economic groups. Our findings are aligned with previous research studies that showed a significant positive linear association between total tax collection, total income, total assets, and total equity during the period of 2015–2019. Furthermore, our analysis discloses that Ecuadorian economic groups tend to compete in oligopolistic markets, given that their economic and financial decisions are interconnected with their family firms or consortium groups. We have detected an exponential growth of the economic groups in Ecuador, in view of the fact that the number of economic groups take off from 17 groups in 2007 to 300 in 2020. Notwithstanding, we predict a stagnation of the economic groups during the post-COVID-19, mainly due to the contraction of 7.8% in the GDP that affects home and government consumption, investment, and exports. For future studies, we recommend conducting an analysis for economic groups, using the industry integration indexes. We would also like to suggest incorporating the labor market characteristics as a crucial variable for the development and analysis of economic policies. Similarly, we recommend focusing on the longitudinal analysis and financial trend indicators for the forthcoming Ecuadorian economic groups. On a final note, and since the period of study employed for this research is from 2015–2019, future researchers can analyze the impact of the concentration of economic power in Ecuador, as well as the impact of the number of economic groups in other countries.

**Author Contributions:** All authors contributed extensively to the work presented in this paper. Writing—original draft preparation, A.B.T.-P.; writing—review and editing, M.E.M.-C. All authors have read and agreed to the published version of the manuscript.

**Funding:** We extend our gratitude and acknowledgment to the Universidad de las Américas, which financially supported this research (2021).

**Data Availability Statement:** The datasets used and analyzed in this study are available from the corresponding author on justified request.

**Conflicts of Interest:** The authors declare no conflict of interest. The funders had no role in the design of the study; in the collection, analyses, or interpretation of data; in the writing of the manuscript, or in the decision to publish the results.

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
