# Peer review of "The Evolution and Takeoff of the Ecuadorian Economic Groups"

_economies, doi:10.3390/economies9040188_

Round 1

Reviewer 1 Report

This is a well written paper with clean but minor contributions, however, qualifies as an appropriate and informative study from Ecuador.

Further I find that COVID-19 as currently mentioned in the title, abstract and conclusion section, is un-necessary and should be removed. This has nothing to do with this study. Nevertheless, authors may discuss post COVID implications or at least expected direction of results if they are expecting strong effects of the COVID on the findings of this study.  

Author Response

Please find the reviewers' response in the attached file.

Reviewer 2 Report

The idea of the study is interesting.

But, several issues are making necessary a revision:

Authors must clearly specify the working hypotheses and methods used to verify them.

Paper lacks a precise indication of the novelty aspect. The authors should point out and highlight the importance of the study.

Section 3 needs to be completely revised .... it is important to present the variables studied, and the methods used. The authors must be describing all methods used with a brief description of them. At line 248, the authors must actually specify the indicators studied (financial and fiscal variables).

Lines 262-263 ”This comprehensive methodology is further addressed along each one of the main findings in the following section”. But where? I did not find them in section 4.

Author Response

(The authors gave the same response as above.)

Round 2

Reviewer 2 Report

The authors revised the article detailedly according to my suggestions, and thus I think the paper improved and can be published.